# Protocol for the MicroRESUS study: The impact of circulatory shock and resuscitation on microcirculatory function and mitochondrial respiration after cardiovascular surgery

John C. Greenwood[1,2]*, Fatima M. Talebi[2], David H. Jang[2], Audrey E. Spelde[3], Todd J. Kilbaugh[4], Frances S. Shofer[5], Michael A. Acker[6], John G. T. Augoustides[3], Jan Bakker[7], Nuala J. Meyer[8], Jacob S. Brenner[8], Vladimir R. Muzykantov[9], Benjamin S. Abella[2]

1 Division of Critical Care Medicine, Department of Emergency Medicine, Department of Anesthesiology and Critical Care, Center for Resuscitation Science, Perelman School of Medicine at the University of Pennsylvania, Philadelphia, PA, United States of America, 2 Department of Emergency Medicine, Center for Resuscitation Science, Perelman School of Medicine at the University of Pennsylvania, Philadelphia, PA, United States of America, 3 Department of Anesthesiology and Critical Care, Perelman School of Medicine at the University of Pennsylvania, Philadelphia, PA, United States of America, 4 Department of Anesthesiology and Critical Care Medicine, Center for Mitochondrial and Epigenomic Medicine, The Children's Hospital of Philadelphia, Philadelphia, PA, United States of America, 5 Department of Epidemiology & Biostatistics, Department of Emergency Medicine Hospital of the University of Pennsylvania, Philadelphia, PA, United States of America, 6 Division of Cardiovascular Surgery, Department of Surgery, Perelman School of Medicine at the University of Pennsylvania, Philadelphia, PA, United States of America, 7 Division of Pulmonary, Allergy, and Critical Care Medicine, New York University, New York, NY, United States of America, 8 Division of Pulmonary and Critical Care, Department of Medicine, Perelman School of Medicine, Philadelphia, PA, United States of America, 9 Department of Pharmacology, University of Pennsylvania, Philadelphia, PA, United States of America

* john.greenwood@pennmedicine.upenn.edu

**Data Availability Statement:** Deidentified microcirculation pilot data are uploaded to the open

## Abstract

### Background

Despite current resuscitation strategies, circulatory shock and organ injury after cardiac surgery occur in 25–40% of patients. Goal-directed resuscitation after cardiac surgery has generated significant interest, but clinical practice to normalize hemodynamic variables including mean arterial pressure, cardiac filling pressures, and cardiac output may not reverse microcirculation abnormalities and do not address cellular dysoxia. Recent advances in technology have made it possible to measure critical components of oxygen delivery and oxygen utilization systems in live human tissues and blood cells. The MicroRESUS study will be the first study to measure microcirculatory and mitochondrial function in patients with circulatory shock and link these findings with clinical outcomes.

### Methods and analysis

This will be a prospective, observational study that includes patients undergoing elective cardiovascular surgery with cardiopulmonary bypass (CPB). Microcirculation will be

access Zenodo database (https://doi.org/10.5281/zenodo.6427023).

**Funding:** JCG is supported by the National Center for Advancing Translational Sciences of the National Institutes of Health, award number KL2TR001879. DHJ is supported by the National Heart, Lung, and Blood Institute of the National Institutes of Health, award number K08HL136858. The content is solely the responsibility of the authors and does not necessarily represent the official views of the National Institutes of Health. Website: https://www.nih.gov/grants-funding The sponsors will not play any role in the study design, data collection, analysis, decision to publish, or preparation of the manuscript.

**Competing interests:** The authors have declared that no competing interests exist.

**Abbreviations:** CABG, Coronary artery bypass graft; C1, Complex 1; CII, Complex II; CIII, Complex III; CIV, Complex IV; CI, Cardiac index; CO, Cardiac output; CPB, Cardiopulmonary bypass; CRF, Case report form; CVP, Central venous pressure; $DO_2$, Rate of oxygen delivery; ERAS, Early recovery after surgery; euroSCORE II, European System for Cardiac Operative Risk Evaluation; IDF, Incident dark field; LFTs, Liver function tests; MAP, Mean arterial pressure; MFI, Microcirculatory flow index; MHI, Microcirculatory heterogeneity index; PAP, Pulmonary artery pressure; PBMCs, Peripheral blood mononuclear cells; PPV, Proportion of perfused vessels; PVD, Perfused vessel density; RBCs, Red blood cells; SOFA, Sequential organ failure assessment; STS, Society of Thoracic Surgeons; SUIT, Substrate-uncoupler-inhibitor titration; $SvO_2$, Central venous oxygen saturation; TVD, Total vessel density; $VO_2$, Rate of oxygen consumption; VVFDs, Ventilator and vasopressor-free days.

quantified with sublingual incident dark field videomicroscopy. Mitochondrial respiration will be measured by performing a substrate–uncoupler–inhibitor titration protocol with high resolution respirometry on peripheral blood mononuclear cells at baseline and serial timepoints during resuscitation and at recovery as a possible liquid biomarker. Plasma samples will be preserved for future analysis to examine endothelial injury and other mechanisms of microcirculatory dysfunction. Thirty-day ventilator and vasopressor-free days (VVFDs) will be measured as a primary outcome, along with sequential organ failure assessment scores, and other clinical parameters to determine if changes in microcirculation and mitochondrial respiration are more strongly associated with clinical outcomes compared to traditional resuscitation targets.

## Discussion

This will be the first prospective study to examine both microcirculatory and mitochondrial function in human patients with circulatory shock undergoing cardiac bypass and address a key mechanistic knowledge gap in the cardiovascular literature. The results of this study will direct future research efforts and therapeutic development for patients with shock.

## Introduction

Despite current resuscitation practices, circulatory shock and perioperative organ injury after cardiac surgery occur in 25–40% of patients [1, 2]. Traditionally, causes of shock after cardiovascular surgery with cardiopulmonary bypass are classified by macrocirculatory derangements (e.g. low cardiac output, vasoplegia, hypovolemia) [3]. Interventions to normalize hemodynamic variables, including mean arterial pressure, cardiac filling pressures, and cardiac output restore the large vessel (macrocirculatory) pressure and flow targets but may not reverse underrecognized disruptions in microcirculatory blood flow and oxygen utilization. Current methods to estimate the balance of oxygen delivery ($DO_2$) relative to demand ($VO_2$) include blood gas-derived calculations and the measurement of downstream biomarkers of anaerobic metabolism such as blood lactate. Using these methods, previous literature has concluded that lactic acidosis after cardiac surgery is unlikely related to inadequate oxygen delivery [4]. Unfortunately these inferences fail to consider the presence of regional blood flow derangements caused by pathologic microcirculatory heterogeneity, which are also associated with severity of postoperative lactic acidosis and organ injury [5, 6]. To resolve this important clinical discrepancy, a deeper understanding of the determinants of oxygen transport pathways and oxygen utilization during health, shock, and resuscitation are needed.

The microcirculation is composed of a network of vessels including arterioles, capillaries, and venules <100 μm in diameter where red blood cells (RBCs), leukocytes, and plasma components interface with the vascular endothelium to allow metabolic substrate exchange. Changes in microcirculatory blood flow can be caused by inflammatory-mediated vascular endothelial injury, microthrombosis, or an inadequate balance between vasoconstrictive and vasodilating agents leading to a global or heterogeneous reduction in capillary blood flow [7, 8]. Incident dark field (IDF) videomicroscopy is a novel, handheld method that can directly image the human microcirculation in real time. Current generation IDF videomicroscopy has improved imaging resolution compared to previous generation devices, and can detect up to 30% more capillaries compared to side stream imaging [9, 10]. As a result, important determinants of tissue oxygenation, such as microcirculatory diffusive and convective properties, can now be more accurately quantified.

Blood gas derived calculations of $VO_2/DO_2$ balance may provide false clinical reassurance as they cannot identify microvascular injury, microvascular shunting, and other mechanisms of dysoxia which may contribute to cell injury and organ dysfunction in patients with shock [11]. Advancements in high resolution respirometry now make it possible to quantify mitochondrial respiration rapidly and reliably in live tissues [12]. Nucleated blood cells (platelets and peripheral blood mononuclear cells) are readily accessible and can be used as surrogates to study cellular respiration in acute care illnesses such as acute heart failure, hemorrhagic shock, sepsis, and patients with ischemic reperfusion injury [13–15]. It is unclear if deficiencies in oxygen delivery and utilization occur independently or concomitantly in patients with shock. Studies that simultaneously examine mitochondrial respiration and microcirculatory function, which are also tied to clinical outcomes, are vitally important to guide future research efforts in therapeutic development and perform effective interventional trials.

## Materials and methods

### Study objective

The primary objective of the MicroRESUS study is to evaluate the microcirculatory and mitochondrial function in patients following elective cardiovascular surgery with cardiopulmonary bypass, to determine if these parameters outperform traditional biomarkers and global hemodynamic measurements to predict clinical outcomes.

### Study design and setting

This is a prospective, observational, single center study with repeated measures from baseline through 30-days after surgery at the University of Pennsylvania in Philadelphia, PA, USA.

### Patient screening

Patients will be screened 24–48 hours prior to surgery using the published operative schedule. First-case elective coronary artery bypass graft (CABG) or valvular surgeries will be considered for study enrollment, pending availability of research personnel to complete the study protocol. Efforts will be made to diversify subject enrollment to accurately reflect the general cardiovascular patient population.

### Informed consent

Patients will be approached for consent by telephone one day prior to surgery or in person while in the preoperative area on the day of surgery. Consent will be obtained by the Principal Investigator or trained research personnel. Patient signatures will be obtained digitally via the REDcap data management system [16]. A digital copy of the consent form will be sent to the patient in addition to being retained by the study team.

### Sample size and power

Using data from our previous foundational work as well as unpublished pilot data, we anticipate a 1:1 allocation of patients to the high ($PVD > 22mm/mm^2$) and low group ($PVD \leq 22$ $mm/mm^2$). We will need a sample size of at least 134 subjects to detect at least a 2-day difference in ventilator and vasopressor-free days (VVFDs) with a $\beta = 0.8$, using a one-sided t-test $\alpha = 0.05$ [5, 6]. We will enroll a total of 140 subjects to allow for a 5% loss to follow-up (surgical delay, ICU delay, cancelled surgery, etc.) and exclusion due inadequate microcirculation video quality (S1 Appendix).

### Inclusion criteria

Adult patients ($\geq$18 years old) receiving elective CABG or valvular surgery requiring cardio-pulmonary bypass are eligible for enrollment. Post-operative patients with circulatory shock will be identified by having:

1. Either vasopressor-dependent hypotension or low cardiac output requiring inotropic support

2. Signs of end-organ injury or impaired tissue perfusion defined by one of the following criteria:

   a. Normothermic patients with a capillary refill time > 3 seconds

   b. Serum lactate > 2 mmol/dL

   c. Mixed venous oxygen saturation (SvO2) < 60%

### Exclusion criteria

Patients will be excluded if they are unable to tolerate sublingual microcirculatory flow imaging (e.g., non-intubated patients dependent upon oxygen by facemask, poor mouth opening), receiving an emergent procedure, have an active malignancy, or mitochondrial disorder.

### Timeline and techniques for data collection

Data for each subject will be collected at time points outlined in Fig 1. Prior to surgery, baseline biologic samples, microcirculation imaging, and clinical data will be obtained. Repeated measurements will be obtained in the ICU and surgical floor. Long-term clinical outcomes will be recorded at 30-days or upon discharge.

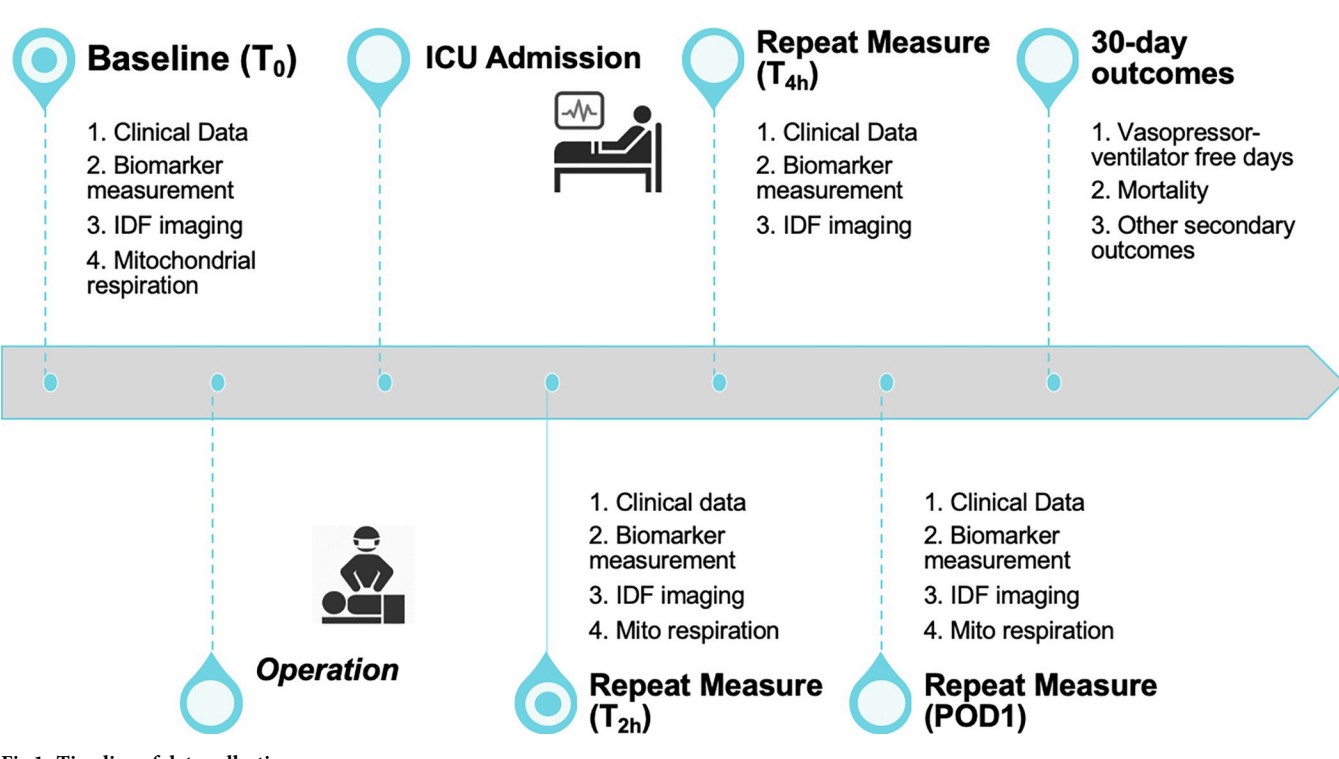

**Fig 1. Timeline of data collection.**

## Demographic and resuscitation data

Demographic values including age, gender, and ethnicity will be recorded. Preoperative risk scores including the STS (Society of Thoracic Surgeons) mortality score and euroSCORE II (European System for Cardiac Operative Risk Evaluation) will be calculated. Intraoperative data including cardiopulmonary bypass time, cross clamp time, blood product use, intravenous fluid administration, and vasoactive administration will be recorded. ICU clinical data (hemodynamics, laboratory testing, etc.), resuscitation data (intravenous fluids, vasoactive administration, blood transfusion, etc.), and clinical outcomes (VVFDs, ICU length of stay, hospital LOS, etc.) will be recorded as indicated in Fig 1. Sequential organ failure assessment (SOFA) scores will be calculated prior to surgery (baseline), as well as 24 and 72 hours after surgery.

## Physiologic and pharmacologic data

Systemic hemodynamic data, perfusion data, and blood gas measurements will be collected upon ICU admission, then hourly during the first 6 hours of postoperative care. Cardiac output (CO), cardiac index (CI), central venous pressure (CVP), pulmonary artery pressure (PAP), and mixed venous oxygen saturation ($SvO_2$) will be monitored continuously using a pulmonary artery catheter (Edwards Lifesciences LLC, Irvine, CA, USA). Arterial blood pressure will be measured using a standard invasive arterial line. Intraoperative and post-operative administration of blood products, intravenous fluids, vasoactive agents (vasopressor and inotropes), and sedatives will be recorded at each time point.

## Incident dark field microscopy

Sublingual microcirculation imaging will be performed using handheld incident dark field (IDF) videomicroscopy (CytoCam, Braedius Medical BV, the Netherlands) at four time points during the enrollment period. Imaging will be performed by the Principal Investigator or trained research personnel. Video sequences are obtained by placing the CytoCam device should sublingual space and maneuvered so that pressure and motion do not result in image artifact (Fig 2). A series of successive video clips (3–5 clips of at least 120 frames or 6 seconds

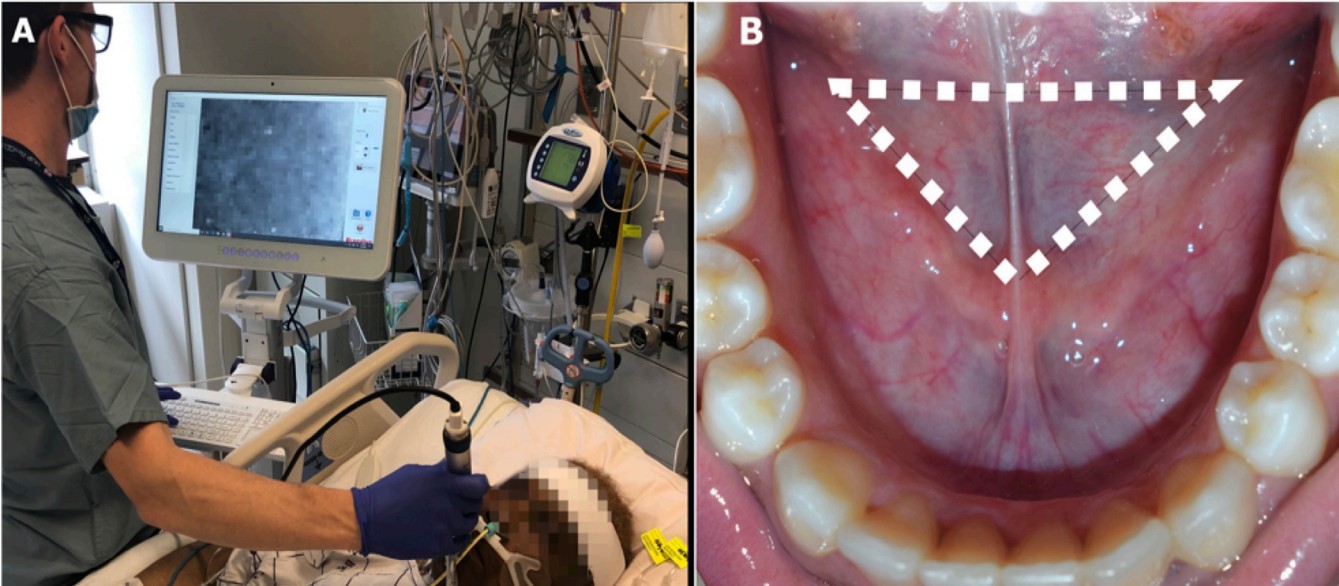

**Fig 2.** A. Experimental setup with patient in supine position during IDF measurement. B. Anatomical sublingual triangle where measurements are obtained.

in length) should be captured in distinct areas of the sublingual space to account for vessel heterogeneity. Focus and lighting during video capture may be adjusted to optimize image acquisition. Baseline ($T_{0h}$) imaging will be obtained in the preoperative area prior to surgery on the day of the scheduled operation. Repeated measurements will be obtained upon arrival to the ICU (0–2 hours after surgery, $T_{2h}$), during ongoing resuscitation (2–4 hours post-op, $T_{4h}$), and on post-operative day one after recovery ($T_{24h}$).

## Analysis of IDF videomicroscopy

Prior to analysis, video quality will be assessed using the 6-factor Massey quality score, which uses a semiquantitative assessment of each video for appropriate illumination, duration, focus, content, stability, and pressure. Only videos with Massey scores of <10 will be included for further analysis [17]. Three videos with the best quality score will be selected for further processing. All IDF images will be coded then analyzed using an offline, dedicated software (Automated Vascular Analysis v3.02, Microvision Medical, The Netherlands). Microvascular flow index (MFI), microcirculatory heterogeneity index (MHI), total vessel density (TVD), proportion of perfused vessels (PPV), and perfused vessel density (PVD). Individual microvessel flow will be scored as 0 = no flow, 1 = intermittent flow, 2 = sluggish flow, or 3 = continuous flow. Vessels will be considered perfused if they are scored as either sluggish or continuous flow. To ensure only vessels contributing to tissue gas exchange and metabolism are included, only vessels < 20 μm in diameter will be analyzed. This process follows the current standard for microcirculation measurement and analysis [18]. Interobserver variation will be tested periodically in at least 10% of the subject videos to ensure minimal scoring heterogeneity between investigators.

## Biological samples

Blood gas samples will be drawn into a commercial, pre-heparinized 1 mL blood sampler then immediately analyzed by an ABL90 FLEX automatic blood gas analyzer (Radiometer America Inc., Brea, California, USA). Arterial blood gas samples will be obtained by clinical staff every 1–2 hours for the first 6 hours after ICU admission based on current clinical practice. Central venous blood gas measurements will also be obtained by the study team at the time of microcirculation measurement. Hemoglobin, hematocrit, pH, $PaO_2$, $PaCO_2$, and glucose are also reported on each blood gas analysis. After 6 hours, blood gas analysis will be performed as needed by the clinical team. For biomarker and mitochondrial function analysis, enrolled patients will undergo phlebotomy with volumes of 15 mL drawn into $K_2EDTA$ tubes. Blood samples will then be centrifuged at room temperature using Ficoll-Paque™ PLUS (GE) and Leucosep tubes (Greiner Bio-one). Plasma specimens will be stored at -80˚C for later evaluation of the mechanisms of microcirculatory dysfunction. Peripheral blood mononuclear cells (PBMCs) will undergo further analysis and processing as detailed below.

## Mitochondrial respiration

A population of PBMCs will be obtained and analyzed from the plasma buffy coat within 1 hour of blood draw. A cell count and viability will be calculated using the Cell Countess II (Invitrogen) with trypan blue exclusion. Between 4–5 x $10^6$ PBMCs will be used for respiration analysis and residual PBMCs will be processed and stored at-80˚C for future quantitative PCR. Unless otherwise specified, all reagents will be obtained from Sigma-Aldrich and Invitrogen.

Mitochondrial respiration will be analyzed using an Oroboros O2k-FluoRespirometer (Oroboros Instruments, Innsbruck, Austria) with a substrate–uncoupler–inhibitor titration (SUIT) protocol and MiR05 buffer [19]. The SUIT protocol measures oxidative

phosphorylation capacity with electron flow through complex I (CI) and complex II (CI + CII) using malate, pyruvate, glutamate, and flavin adenine dinucleotide-linked substrate succinate in the presence of adenosine diphosphate. The addition of digitonin allows for the measurement of specific complex-linked activity. Oligomycin, an inhibitor of the ATP synthase, uncouples respiration from ATP-synthase activity to measure respiration where the $O_2$ consumption is dependent on the leakiness the mitochondrial membrane and back-flux of protons into the mitochondrial matrix independent of ATP synthase ($LEAK_{CI+CII}$). Maximal convergent non-phosphorylating respiration of $ETS_{CI+CII}$ is evaluated by titrating the protonophore, carbonyl cyanide p-(trifluoromethoxy) phenylhydrazone. $ETS_{CI+CII}$ is considered a stress test for mitochondria, a marker of mitochondrial respiratory reserve. Non-phosphorylating respiration specifically through CII ($ETS_{CII}$) is achieved through the addition of rotenone, an inhibitor of CI. The complex III (CIII) inhibitor antimycin-A is added to measure the residual non-mitochondrial oxygen consumption, and this value is subtracted from each of the measured respiratory states to provide only mitochondrial respiration. Complex IV (CIV)-linked respiration will be measured by the addition of ascorbate with N,N,N,N-tetramethyl-phenylenediamine. The CIV inhibitor sodium azide will be added to reveal the chemical background that is subtracted from the N,N,N,N-tetramethyl-phenylenediamine-induced oxygen consumption rate. Mitochondrial reactive oxygen species production will be measured using the Amplex UltraRed method [20]. All data will be acquired using DatLab 7 (Oroboros Instruments, Innsbruck, Austria) and respiration value will be normalized to cell count.

## Outcome measures

Our hypothesis is that patients with poor post-operative sublingual functional capillary density (defined as a PVD < 22 mm/mm$^2$ and MHI > 0.4) will have a higher degree of postoperative cardiovascular and pulmonary organ injury compared to patients with normal postoperative microcirculation. The primary outcome for this study will be VVFDs during the first 30 days after surgery. This outcome was chosen because our elective cardiac surgery patients are managed using early recovery after surgery (ERAS) protocols, with a goal to be extubated and weaned off vasopressors within 24 hours after surgery [21, 22]. VVFDs will be calculated as a reverse count of consecutive days without requiring ICU-level respiratory or vasopressor support (Fig 3). Our research group has used a similar primary outcome in a previous clinical trial [23]. The day of operation (day zero) will not be included as all patients undergoing cardiovascular surgery with CPB require mechanical ventilation and vasoactive medications on the day of surgery. Ventilator days will include mechanical ventilation via endotracheal tube or tracheostomy, high-flow nasal cannula ≥ 40 liters/minute with an FiO2 > 40%, or non-invasive positive pressure ventilation not prescribed at home. Non-invasive positive pressure ventilation and HFNC will be included as these both require the patient to remain in our ICU. Vasoactives include epinephrine, norepinephrine, vasopressin, phenylephrine, dobutamine, or milrinone at any dose. Subjects in need of respiratory or vasopressor support on day 30, or die before day 30 will be assigned zero VVFDs. If the patient is discharged prior to hospital day 30 a "last status carried forward" approach will be used. Secondary outcomes will include sequential organ failure assessment (SOFA) score on day 1, day 3, diagnosis of acute kidney injury (defined as an increse in serum creatinine ≥ 0.3 mg/dL within 2 days), and hospital length of stay.

Exploratory outcomes will examine the relationship of microcirculatory function with common clinical biomarkers such as lactate, SvO2, venous-to-arterial PCO$_2$ gap, capillary refill time, and lactate to pyruvate ratio. Markers of endothelial injury and inflammation such as soluble vascular cell adhesion molecule 1 (sVCAM-1), soluble intercellular adhesion molecule 1 (sICAM-1), e-selectin, IL-8, IL-10, and other inflammatory cytokines will be measured. We

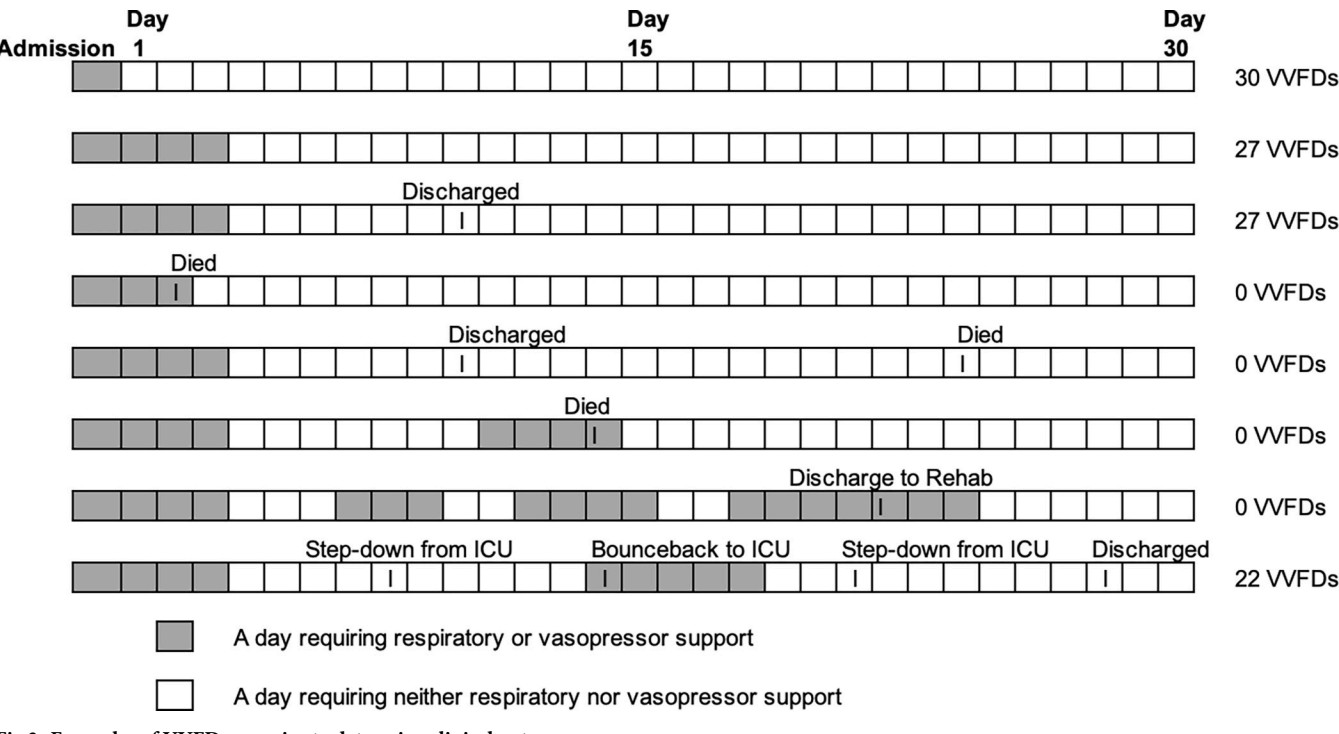

**Fig 3. Examples of VVFD scenarios to determine clinical outcomes.**

will also examine mitochondrial complex function in human blood cells, comparing individual complex (I, II, III, and IV) function and reactive oxygen species generation between patients with and without post-operative shock.

## Safety considerations

The Principal Investigator will be primarily responsible for the study conduction throughout protocol completion. Research personnel will be required to report any relevant protocol deviations or research related adverse events to the Principal Investigator. Risks associated with sublingual IDF imaging are low and no identifiable adverse events have been previously linked to sublingual IDF imaging.

## Statistical and analysis plans

Continuous variables characterizing demographical data, microcirculation data, and mitochondrial respiration measurements, and outcomes data will be reported as means with standard deviations if normally distributed or medians with interquartile ranges if not normally distributed. Categorical variables will be represented as frequencies and proportions. To examine the predictive performance of selected variables for the primary outcome, we will construct receiver operator characteristic curves for threshold values of PVD, MHI, lactate, $SvO_2$, mean arterial pressure, and cardiac index. A Youden index will be calculated to determine the best cutoff value for determining prolonged VVFDs.

Linear regression modeling will be used to examine the relationship between L/P ratio and postoperative microcirculation variables. We will perform univariate analyses on candidate predictor variables of L/P ratio including PVD, MHI, LFTs, creatinine, CPB time, cross clamp time, and catecholamine administration. Multiple linear regression analysis will be used to

model the effect of significant predictors. Repeated measure ANOVA will be used to compare changes in microcirculatory variables, mitochondrial respiration, and mitochondrial reactive oxygen species production over time. To adjust for multiple comparisons, post-hoc pairwise Tukey Kramer t-tests will be performed. All analyses will use statistical software (SAS version 15.1, Cary, NC; Prism v 9.0, Graph-Pad Software, San Diego, CA).

### Data storage and management

All clinical and research related data will be recorded on study specific case report forms (CRFs) kept in the possession of the Principal Investigator. Data will be transferred to the secure HIPAA-compliant online clinical research database tool, REDCap (REDCap, Vanderbilt University, Nashville, TN) [24]. Subject data will be deidentified and only accessible by appropriate research personnel. All biological samples will be identifiable only by study subject code, and after analysis is complete the samples will be disposed.

### Ethics and dissemination

This study is approved by the University of Pennsylvania Institutional Review Board (IRB # 829765) and informed consent will be obtained prior to enrollment. The dataset supporting the results of this study will be available in the Zenodo research data repository. This study is registered with ClinicalTrials.gov at NCT05330676.

### Status and timeline of the study

Initial pilot testing, staff education, and laboratory calibration testing began recruitment of the participants began on September 1, 2020. The study is actively enrolling subjects at the time of this publication. Preliminary analysis of microcirculation images, mitochondrial respiration, and VVFDs will be conducted in 2022.

## Discussion

Goal-directed resuscitation strategies have improved postoperative clinical outcomes over the past two decades, but many patients continue to experience significant morbidity including multiorgan dysfunction after cardiac surgery [25]. The evolution of previous technologies has made it possible to evaluate real-time microcirculatory function and perform rapid analysis of mitochondrial respiration in critically ill adults, which make it possible to explore alternative mechanisms of organ injury not addressed by current resuscitation practices.

It is not fully known if there is a discordance between macrocirculatory hemodynamic targets and microcirculatory blood flow, as this has only been shown in studies with a generally small sample size. Of particular interest will be early differences in microcirculatory and mitochondrial function in patients with and without post-operative shock. In order to improve outcomes of these critically ill patients and prevent unnecessary perioperative morbidity, examination of mechanisms not addressed by standard resuscitation practices must be explored.

We recognize that there may be some limitations to using the sublingual site to estimate global microcirculatory function and PBMCs to estimate of global mitochondrial function in a diverse, clinical patient population. To minimize differences between patients, we plan to match patient groups for comorbidities such as diabetes, coronary artery disease, and medications that could depress mitochondrial function. The sublingual microcirculation is embryologically related to the gastrointestinal tract, and has been correlated with splanchnic circulation which is one of the most significant lactate producers during shock [26].

Additionally, PBMCs have been found to correlate with renal and cardiac mitochondrial function in hemorrhagic shock [14]. Our team has shown that PBMC mitochondrial ROS (mROS) track closely with brain tissue mROS in an ischemia-reperfusion model) [15]. This supports the concept of using sublingual microcirculation and PBMCs to reflect global injury.

The MicroRESUS study will be the first to simultaneously examine two underrecognized, discrete causes of cellular hypoxia in human subjects with shock with consideration of important clinical outcomes. This study will address a critical gap in the literature, by simultaneously measuring microcirculatory and mitochondrial function in patients with shock and tying our findings to important clinical outcomes. Results from this study will guide future research efforts to identify, develop, and design novel therapies and interventional trials to reverse unrecognized physiologic derangements after cardiovascular surgery.

## Supporting information

**S1 Appendix. Power calculations and determination of sample size.**
(DOCX)

## Author Contributions

**Conceptualization:** John C. Greenwood, David H. Jang, Todd J. Kilbaugh, Michael A. Acker, John G. T. Augoustides, Jan Bakker, Nuala J. Meyer, Jacob S. Brenner, Vladimir R. Muzykantov, Benjamin S. Abella.

**Data curation:** John C. Greenwood, Fatima M. Talebi, David H. Jang, Audrey E. Spelde, Todd J. Kilbaugh.

**Formal analysis:** John C. Greenwood, Fatima M. Talebi, Frances S. Shofer, Benjamin S. Abella.

**Funding acquisition:** John C. Greenwood.

**Investigation:** John C. Greenwood, Fatima M. Talebi, Audrey E. Spelde, Todd J. Kilbaugh, Michael A. Acker, John G. T. Augoustides, Jan Bakker, Nuala J. Meyer, Jacob S. Brenner.

**Methodology:** John C. Greenwood, Audrey E. Spelde, Frances S. Shofer, Michael A. Acker, Jan Bakker, Nuala J. Meyer, Jacob S. Brenner, Vladimir R. Muzykantov, Benjamin S. Abella.

**Resources:** John C. Greenwood, Jan Bakker, Nuala J. Meyer, Benjamin S. Abella.

**Supervision:** Todd J. Kilbaugh, Frances S. Shofer, Michael A. Acker, John G. T. Augoustides, Nuala J. Meyer, Vladimir R. Muzykantov, Benjamin S. Abella.

**Writing – original draft:** John C. Greenwood, Fatima M. Talebi, David H. Jang, Audrey E. Spelde.

**Writing – review & editing:** John C. Greenwood, Fatima M. Talebi, David H. Jang, Audrey E. Spelde, Todd J. Kilbaugh, Michael A. Acker, John G. T. Augoustides, Jan Bakker, Nuala J. Meyer, Jacob S. Brenner, Vladimir R. Muzykantov, Benjamin S. Abella.

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
