## [Decision Letter · Decision Letter 0]

18 Jul 2022

PONE-D-22-12392Protocol for the MicroRESUS study: The impact of circulatory shock and resuscitation on microcirculatory function and mitochondrial respiration after cardiovascular surgery.PLOS ONE

Dear Dr. Greenwood,

Thank you for submitting your manuscript to PLOS ONE. After careful consideration, we feel that it has merit but does not fully meet PLOS ONE’s publication criteria as it currently stands. Therefore, we invite you to submit a revised version of the manuscript that addresses the points raised during the review process.

Overall, the manuscript is satisfactory. Several minor and less minor changes are requested by the Reviewers. One point noted by the team was the start date of the study appears to precede the present manuscript by a few years. Is there a reason for this? 

We look forward to receiving your revised manuscript.

Kind regards,

Jeffrey S Isenberg, MD, MPH

Academic Editor

PLOS ONE

Journal Requirements:

“Abramson 412 Emergency Medicine & Critical Care Research Fund”

“JCG is supported by the National Center for Advancing Translational Sciences of the National Institutes of Health, award number KL2TR001879. DHJ is supported by the National Heart, Lung, and Blood Institute of the National Institutes of Health, award number K08HL136858. The content is solely the responsibility of the authors and does not necessarily represent the official views of the National Institutes of Health.

Website: https://www.nih.gov/grants-funding

The sponsors will not play any role in the study design, data collection, analysis, decision to publish, or preparation of the manuscript.”

4. One of the noted authors is a group or consortium “The MicroRESUS Investigator Group”. In addition to naming the author group, please list the individual authors and affiliations within this group in the acknowledgments section of your manuscript. Please also indicate clearly a lead author for this group along with a contact email address.

Additional Editor Comments:

The authors are requested to make a number of changes in the manuscript. Of note, the study began several years ago. Can a reason be provided in the text for why the present manuscript was not developed and submitted sooner?

Reviewers' comments:

Reviewer's Responses to Questions

**Comments to the Author**

1. Does the manuscript provide a valid rationale for the proposed study, with clearly identified and justified research questions?

Reviewer #1: Yes

Reviewer #2: Yes

2. Is the protocol technically sound and planned in a manner that will lead to a meaningful outcome and allow testing the stated hypotheses?

Reviewer #1: Yes

Reviewer #2: Partly

3. Is the methodology feasible and described in sufficient detail to allow the work to be replicable?

Reviewer #1: Yes

Reviewer #2: No

4. Have the authors described where all data underlying the findings will be made available when the study is complete?

Reviewer #1: Yes

Reviewer #2: Yes

5. Is the manuscript presented in an intelligible fashion and written in standard English?

Reviewer #1: Yes

Reviewer #2: Yes

6. Review Comments to the Author

You may also provide optional suggestions and comments to authors that they might find helpful in planning their study.

Reviewer #1: Thank you for this great study protocol article. I only wonder if measurements are also to be done during the operation since predominantly the start of CPB may be associated with microcirculatory derangement (Den Uil et al, J Thorac Cardiovasc Surg 2008). Good luck!

Reviewer #2: The authors provided a study protocol for microcirculatory assessment in patients undergoing cardiac surgery. As the restoration of tissue perfusion is the primary goal of hemodynamic therapy, the study is of high interest. However, I have some questions and concerns about the manuscript and the study design:

1) The title of the study implies that you will investigate patients with shock. However, the inclusion criteria used do not include a marker for shock or hemodynamic instability. I think, only the minority of the included patients will suffer from shock. Please provide how shock will be defined in the study. What percentage of patients in your hospital develop shock during elective cardiac surgery procedures?

2) As primary endpoint you plan to use ventilator and vasopressor-free days. Further, you state the hypothesis that patients with poor post-operative microcirculatory function will have a higher degree of postoperative organ injury compared to patients with normal postoperative microcirculation. If you want to test this hypothesis, why do you not use a score for organ failure as the primary endpoint? Which parameters and cut-off values do you use to differentiate between normal and poor microcirculation?

3) In the statistics section you write "Continuous variables characterizing each study group will be reported as means with standard deviations or medians with interquartile ranges.". Which groups?

4) You used a 2-day difference in ventilator and vasopressor-free days for sample size calculation. Which groups and what group sizes did you assume for this?

7. PLOS authors have the option to publish the peer review history of their article (what does this mean?). If published, this will include your full peer review and any attached files.

Reviewer #1: No

Reviewer #2: No

---

## [Author Response · Author response to Decision Letter 0]

27 Jul 2022

Hello! We are thankful for the opportunity to resubmit our study protocol. We have made edits to our manuscript document and outlined each change in the, "Response to Reviewers" document. Please let me know if you have any additional questions. Thanks!

- John Greenwood

---

## [Decision Letter · Decision Letter 1]

8 Aug 2022

Protocol for the MicroRESUS study: The impact of circulatory shock and resuscitation on microcirculatory function and mitochondrial respiration after cardiovascular surgery.

PONE-D-22-12392R1

Dear Dr. Greenwood,

We’re pleased to inform you that your manuscript has been judged scientifically suitable for publication and will be formally accepted for publication once it meets all outstanding technical requirements.

Kind regards,

Jeffrey S Isenberg, MD, MPH

Academic Editor

PLOS ONE

Additional Editor Comments (optional):

The Reviewers found that the revised manuscript draft addressed their concerns. The authors are thanked for the additional effort.

Reviewers' comments:

Reviewer's Responses to Questions

**Comments to the Author**

1. Does the manuscript provide a valid rationale for the proposed study, with clearly identified and justified research questions?

Reviewer #1: Yes

Reviewer #2: Yes

2. Is the protocol technically sound and planned in a manner that will lead to a meaningful outcome and allow testing the stated hypotheses?

Reviewer #1: Yes

Reviewer #2: Yes

3. Is the methodology feasible and described in sufficient detail to allow the work to be replicable?

Reviewer #1: Yes

Reviewer #2: Yes

4. Have the authors described where all data underlying the findings will be made available when the study is complete?

Reviewer #1: Yes

Reviewer #2: Yes

5. Is the manuscript presented in an intelligible fashion and written in standard English?

Reviewer #1: Yes

Reviewer #2: Yes

6. Review Comments to the Author

You may also provide optional suggestions and comments to authors that they might find helpful in planning their study.

Reviewer #1: Thank you for your answer to my comment.

In my opinion, the authors satisfactorily answered my question.

Reviewer #2: Microcirculatory monitoring is not available in clinical routine so far. Therefore the study is of high interest. The methody have been sufficiently described to replicate the investigation. The revision improved the manuscript, so the objectives of the study are now comprehensible and achievable with the methods used. Thank you my questions have been answered adequately.

7. PLOS authors have the option to publish the peer review history of their article (what does this mean?). If published, this will include your full peer review and any attached files.

Reviewer #1: No

Reviewer #2: No

---

## [Editor Report · Acceptance letter]

11 Aug 2022

PONE-D-22-12392R1 

Protocol for the MicroRESUS study: The impact of circulatory shock and resuscitation on microcirculatory function and mitochondrial respiration after cardiovascular surgery. 

Dear Dr. Greenwood:

I'm pleased to inform you that your manuscript has been deemed suitable for publication in PLOS ONE. Congratulations! Your manuscript is now with our production department. 

Kind regards, 

on behalf of

Dr. Jeffrey S Isenberg 

Academic Editor

PLOS ONE